# A Multi-Omics Approach to Disclose Metabolic Pathways Impacting Intestinal Permeability in Obese Patients Undergoing Very Low Calorie Ketogenic Diet

**DOI:** 10.3390/nu16132079

**Published:** 2024-06-28

**Authors:** Giuseppe Celano, Francesco Maria Calabrese, Giuseppe Riezzo, Benedetta D’Attoma, Antonia Ignazzi, Martina Di Chito, Annamaria Sila, Sara De Nucci, Roberta Rinaldi, Michele Linsalata, Carmen Aurora Apa, Leonardo Mancini, Maria De Angelis, Gianluigi Giannelli, Giovanni De Pergola, Francesco Russo

**Affiliations:** 1Department of Soil, Plant and Food Science, University of Bari Aldo Moro, 70126 Bari, Italy; giuseppe.celano@uniba.it (G.C.); carmen.apa@uniba.it (C.A.A.); leonardo.mancini1@uniba.it (L.M.); maria.deangelis@uniba.it (M.D.A.); 2Functional Gastrointestinal Disorders Research Group, National Institute of Gastroenterology IRCCS “Saverio de Bellis”, 70013 Castellana Grotte, Italy; giuseppe.riezzo@irccsdebellis.it (G.R.); benedetta.dattoma@irccsdebellis.it (B.D.); antonia.ignazzi@irccsdebellis.it (A.I.); michele.linsalata@irccsdebellis.it (M.L.); 3Center of Nutrition for the Research and the Care of Obesity and Metabolic Diseases, National Institute of Gastroenterology IRCCS “Saverio de Bellis”, 70013 Castellana Grotte, Italy; dichitomartina@gmail.com (M.D.C.); annamaria.sila@irccsdebellis.it (A.S.); sara.denucci@irccsdebellis.it (S.D.N.); roberta.rinaldi@irccsdebellis.it (R.R.); giovanni.depergola@irccsdebellis.it (G.D.P.); 4Scientific Direction, National Institute of Gastroenterology IRCCS “Saverio de Bellis”, 70013 Castellana Grotte, Italy; gianluigi.giannelli@irccsdebellis.it

**Keywords:** metabolomics, calorie restrictive diet, VOCs, dietary intervention, obesity, 16S microbiota profile

## Abstract

A very low calorie ketogenic diet (VLCKD) impacts host metabolism in people marked by an excess of visceral adiposity, and it affects the microbiota composition in terms of taxa presence and relative abundances. As a matter of fact, there is little available literature dealing with microbiota differences in obese patients marked by altered intestinal permeability. With the aim of inspecting consortium members and their related metabolic pathways, we inspected the microbial community profile, together with the set of volatile organic compounds (VOCs) from untargeted fecal and urine metabolomics, in a cohort made of obese patients, stratified based on both normal and altered intestinal permeability, before and after VLCKD administration. Based on the taxa relative abundances, we predicted microbiota-derived metabolic pathways whose variations were explained in light of our cohort symptom picture. A totally different number of statistically significant pathways marked samples with altered permeability, reflecting an important shift in microbiota taxa. A combined analysis of taxa, metabolic pathways, and metabolomic compounds delineates a set of markers that is useful in describing obesity dysfunctions and comorbidities.

## 1. Introduction

The prevalence of obesity is expected to reach 50% worldwide by 2035, and it is mainly attributed to a combination of genetic predispositions and environmental factors, rendering it a complex chronic condition. Given its crucial connections with dietary habits, gut microbiome, and fecal/urinary metabolome, a comprehensive understanding of its etiology is crucial. This complexity underscores significant health ramifications, including heightened susceptibilities to coronary heart disease, hypertension, diabetes mellitus, gallbladder disease, osteoarthritis, and specific forms of cancer [1].

The common denominator of obesity and some of its related comorbidities is a sub-inflammatory status, featuring the secretion/release of pro-inflammatory cytokines from adipose tissue and a consequent infiltration of leukocytes and macrophages into the adipose tissue [2]. The persistent inflammation leads to an increased gut epithelial permeability sustained by an unregulated transit of lipopolysaccharide (LPS) and other inflammatory triggers, directly released into the bloodstream and leading to the assembly of multimeric signaling complexes called inflammasomes [3]. Beyond the high presence of pro-inflammatory markers, one of the major risk factors consists of the intake of nutrients from westernized diets that alone can lead to a shift in the gut microbiome profile and can negatively impact the intestinal permeability [4].

Among various suggested dietary interventions, very low calorie ketogenic diets (VLCKDs) have proven effective for weight loss in obese individuals, affecting the associated metabolic processes. However, a VLCKD’s potential impacts on gut microbiota composition, function, and gut barrier integrity may raise concerns. A VLCKD could exacerbate an already compromised intestinal balance, such as that in obese individuals. Our previous investigations suggested a correlation between the integrity of the intestinal barrier, the ketogenic diet, and its influence on the gut microbiota in obese patients. Our study revealed that obese patients following an 8-week VLCKD treatment had a significant reduction in body weight but exhibited impaired intestinal permeability, dysbiosis, and increased serum levels of LPS (lipopolysaccharides). However, our obese patients did not respond uniformly to VLCKD treatment, which might trigger an intestinal barrier impairment [5]. As indicated by an increased lactose/mannose ratio, this difference can be attributed to a shift from glucose to ketone bodies.

In the analyses shown here, beyond the VLCKD effect on energy consumption, we report how this dietary regimen impacts gut microbiota taxa composition and density depending on intestinal permeability. In line with an increasing body of evidence stating a strict association between gut microbiota dysbiosis and obesity comorbidities, we better detailed a more conspicuous reshaping of microbiota metabolism in VLCKD obese patients with altered permeability when compared with those marked by a normal permeability status.

## 2. Materials and Methods

### 2.1. Study Experimental Design and Recruited Participants

Our drawn conclusions are based on the experimental design carried out at the Italian Centre of Nutrition for the Research and Care of Obesity and Metabolic Diseases, partner of the National Institute of Gastroenterology IRCCS “Saverio de Bellis” in Castellana Grotte (Ba).

Recruited obese participants had an age ranging from 18 to 65 years and a BMI exceeding 30 kg/m^2^. Medical history review, physical examination, and laboratory tests were performed on the entire cohort that was composed of 25 patients without irritable bowel syndrome (IBS). Based on the lactulose/mannitol ratio, this cohort was further subdivided into two sub-clusters, consisting of 14 samples indicating normal permeability and 11 samples indicating altered permeability. Fecal and urinary samples were collected in three days before starting the VLCKD treatment, as reported in our previous study [6].

During the collection of participants’ medical history, information regarding smoking and daily alcohol consumption habits was obtained. Specifically, participants were asked to adhere to the American and European guidelines, indicating whether they consumed more than two glasses of alcohol per day (or one for females) [6]. The threshold for men was set at 30 g/day, while for women, it was 20 g/day.

Exclusion criteria encompassed hypersensitivity to meal replacement, patients suffering from cerebrovascular and cardiac diseases, respiratory insufficiency, type 1 diabetes mellitus, severe gastrointestinal diseases, chronic kidney disease, pregnancy, lactation, and psychiatric issues. Additionally, we took into consideration the presence of eating disorders, liver failure, substance abuse, frail elderly patients, active/severe infections, rare diseases, disorders related to mitochondrial fatty acid oxidation, and serious mental illnesses. Moreover, 15 days before starting the dietary intervention, participants were asked to stop any probiotics use, together with vitamins, drugs, or any other additional supplement.

The used protocol (Prot. n. 170/CE De Bellis) received approval from the internal Medical Ethical Committee and respected the Helsinki Declaration (1964). Participants signed a written consent form before their enrolment. The present study is presented within the ClinicalTrials.gov database and corresponds to the identifier NCT05477212.

The analyses carried out in the present paper are based on the sample cohort stratification that relies on the assessment of intestinal integrity. More in detail, as reported in the previous pilot study [5], the integrity of intestinal barrier was assessed based on the Lac/Man ratio, using a threshold value of 0.03.

### 2.2. DNA Extraction from Fecal Samples

Fresh fecal samples were collected from a total of 20 obese subjects before and after being administered with the VLCKD. With respect to the total number of samples included in the clinical trial (n = 25), five samples were dropped out because of the poor quality of extracted fecal DNA. Among these 40 sample points, 1 sample gave too few reads to be compared and was dropped out. Thus, a total of 19 samples were kept in the whole batch, as reported in the CONSORT flowchart (Appendix A).

Total bacterial DNA was extracted from stool samples by means of the QIAamp FAST DNA Stool Mini Kit (Qiagen, Hilden, Germany), according to the manufacturer’s instructions; a NanoDrop ND-1000 spectrophotometer (Thermo Scientific, Waltham, MA, USA) and Qubit Fluorometer 1.0 (Invitrogen Co., Carlsbad, CA, USA) were used to determine the yield and quality of extracted DNA. The 16S metagenomic sequencing was in service performed at Genomix4life S.R.L. (Baronissi, Salerno, Italy). More specifically, the amplification of the V3 and V4 regions of the 16S target gene was obtained by using the following couple of primers, forward: 5′-CCTACGGGNGGCWGCAG-3′ and reverse: 5′-GACTACHVGGGTATCTAATCC-3′. Metagenomic Sequencing Library Preparation (Illumina, San Diego, CA, USA) was used as a guide reference for PCR reactions. A negative control was added to the experiment in order to avoid contamination. The resulting libraries were then quantified using a Qubit fluorometer (Invitrogen Co., Carlsbad, CA, USA) and pooled to an equimolar amount of each index-tagged sample at a final concentration of 4 nM, including the Phix Control Library. The pooled samples were subjected to cluster generation and sequenced on the MiSeq platform (Illumina, San Diego, CA, USA) in a 2 × 300 paired-end format.

### 2.3. Sequencing Quality Check and Metataxonomic Bioinformatics Pipeline

Raw reads’ quality was checked by FastQC software. From denoising-to-taxonomic assignment, in silico bioinformatics analyses were performed in a “anaconda” environment ad hoc customized on the QIIME2 [7] microbiome platform (version 2020.8) and relative plugins. More in depth, QIIME plugin q2-deblur (https://github.com/qiime2/q2-deblur, accessed on 25 March 2024) was specifically used in the denoising step, and alpha (Shannon entropy and Faith’s PD) and beta diversity metrics, together with the relative statistics (PERMANOVA), were run using a nested function. The database used for metabarcoding annotation was SILVA 138 (https://www.arb-silva.de/documentation/release-138/, accessed on 25 March 2024) to infer relative taxonomy.

Statistically significant changes in alpha and beta diversity were assessed by Bray–Curtis, Jaccard, Weighted Unifrac, and Unweighted Unifrac distance matrices computed using QIIME II nested plugins. The same software was used to compute PERMANOVA test. White’s non-parametric statistic corrected for multiple tests (Benjamini–Hochberg) allowed us to retrieve statistically significant hits in the two groups’ pairwise comparisons that where then plotted as extended error bar plots, thanks to the STAMP 8.30 software python routine environment. More in detail, the plot reports the effect size and associated confidence interval for each significative feature (corrected *p* < 0.05) obtained in STAMP software by applying corrected Welch test (BH) statistics based on the normalized matrix from QIIME2. The scale value reports the relative abundances transformed in mean proportions. The software computes the mean proportions that are taken over by the whole set of samples belonging to each group.

### 2.4. Metabolic Pathway Prediction

Phylogenetic Investigation of Communities by Reconstruction of Unobserved States (PICRUSt2) pipeline version 2.0 (https://picrust.github.io/picrust/, accessed on 25 March 2024) was run as an extension of QIIME2 software in the same anaconda environment. The MetaCyc pathway abundance matrix was directly employed as input for comparing the groups.

### 2.5. NCBI Bioproject

Sequencing fastQ raw files were used to create a dedicated NCBI bioproject for VLCKD samples (PRJNA1090658).

### 2.6. Fecal and Urinary GC-MS Metabolite Profiles

An experiment of gas chromatography coupled with mass spectrometry was performed on fecal and urinary samples. In detail, one gram of feces (*n* = 29 and *n* = 14 for the probiotic and placebo groups, respectively) was placed in 10 mL glass vials. Ten microliters of 4-methyl-2-pentanol (final concentration of 1 mg L^−1^) were added as an internal standard. Samples were equilibrated for 10 min at 60 °C. Solid-phase microextraction fiber (divinylbenzene/Carboxen/polydimethylsiloxane) was exposed to each sample for 40 min. The VOCs were thermally desorbed by immediately transferring the fiber to the heated injection port (220 °C) of Clarus 680 (Perkin Elmer, Beaconsfield, UK) gas chromatography equipped with an Rtx-WAX column (30 m × 0.25 mm i.d.; 0.25 µm film thickness) (Restek) and coupled to a Clarus SQ8MS (Perkin Elmer) [6]. The column temperature was set initially at 35 °C for 8 min, and then it was increased to 60 °C at 4 °C min^−1^, to 160 °C at 6 °C min^−1^, and finally to 200 °C at 20 °C min^−1^ and held for 15 min. Urine was collected individually in safe, sterile boxes.

A 20 mL glass vial was supplied with 2 g urine plus 10 μL of internal standard solution (2-pentanol-4-methyl) at 33 ppm. Vials were sealed with polytetrafluoroethylene-coated silicone rubber septa (20-mm diameter; Supelco, Bellefonte, PA, USA). To obtain the best extraction efficiency, the solid-phase microextraction (SPME) was performed by exposing a conditioned 75 μm Carboxen/PDMS fiber (Supelco, Bellefonte, PA, USA) to the headspace of 2 mL of acidified (pH 2) urine sample with 1 g of NaCl for 60 min at 60 °C after a 35 min incubation) [6]. The extracted compounds were desorbed in splitless for 3 min at 280 °C. A Clarus 680 (PerkinElmer, Waltham, MA, USA) gas chromatograph equipped with an Elite-624Sil MS Capillary Column (30 m × 0.25 mm i.d.; 1.4 μm film thickness; PerkinElmer) was used. The column temperature was set initially at 40 °C for 3 min and then increased to 250 °C at 5 °C/min and to 280 °C at 10 °C/min and finally held for 5 min. Helium was used as the carrier gas at a flow rate of 1 mL/min. The analyses lasted 58 min.

Spitless injection was used for sample introduction into the capillary column. Helium was used as the carrier gas, with a flow rate of 1 mL min^−1^. The source and transfer line temperatures were maintained at 250 °C and 230 °C, respectively. Electron ionization masses were recorded at 70 eV in the mass-to-charge ratio interval, which was from 34 to 350 *m*/*z*. The gas chromatography–mass spectrometry generated a chromatogram with peaks representing individual compounds. Each chromatogram was analyzed for peak identification using the National Institute of Standard and Technology 2008 library. A peak area threshold of >1,000,000 and 85% or greater probability of match was used for VOC identification, followed by manual visual inspection of the fragment patterns when required. 4-Methyl-2-pentanol (final concentration 1 mg L^−1^) was used as an internal standard in all analyses to quantify the identified compounds via interpolation of the relative areas versus the internal standard area.

### 2.7. Statistical Analyses

Multivariate statistical analyses inclusive of principal component analysis and partial least square differential analysis (PLS-DA) were run in R environment, using the “prcomp” and “PLSR.Anal” functions implemented in R MetaboAnalystR-software (version 2.0.0), respectively. The VIP score indicated those variables that most contributed to the PLS-DA clustering. A non-parametric Wilcoxon rank-sum test combined with a fold-change analysis was rendered as a volcano plot, which was useful in summarizing the statistically significant metabolic pathways.

Pairwise group taxa comparisons were performed based on the White’s non-parametric statistical test corrected for multiple tests (Benjamini–Hochberg), and only statistically significant hits (corrected *p* < 0.05) were further inspected. Relative results were then plotted as extended error bar plots, thanks to the STAMP software python routine environment. In all the considered analyses, only multiple-test-corrected items were considered (q < 0.05).

## 3. Results

We here primarily investigated the microbiota taxa differences (16S rRNA gene sequencing) emerging from VLCKD-administered subjects with normal and altered intestinal permeability. Our step-by-step investigation protocol, including four papers, moved from the investigation of clinical parameter changes in the obese patient cohort under VLCKD [5] to the evaluation of metabolite signatures in two different stratifications based on treatment and intestinal permeability [6,8]. The 16S taxa profiles were evaluated by stratifying patients based on the same stratifications.

The present sequencing metataxonomic dataset was evaluated by tracing a crosslink between VOC profiles derived from both fecal and urine samples in paired subjects. The complete panel of 16S sequencing-derived taxa was inspected in terms of both alpha and beta diversity, as well as in terms of statistically significant predicted pathways based on microbiota taxa abundances.

### 3.1. Multivariate Analyses Based on Single and Mixed Data Matrices

Upon the first data inspection, the supervised clustering approach did not resolve clouds relative to samples from subjects with altered permeability. The PLS-DA plot based on taxa at the genus level failed in distinguishing samples belonging to the four assigned groups (Appendix A).

The same multivariate analysis takes advantage of the merging together of variables from 16S rRNA analyses and untargeted metabolomics both from fecal and urines samples (Figure 1 and Appendix A). Although with a higher background noise, the samples were distinguished in four groups.

Looking at variables with a higher impact (high VIP score), we see that the genus *Butyrricicoccus* was the one that had the highest VIP score and was specifically associated with the post-treated group with altered permeability. Moreover, among the top VIPs with a score greater than 2.3, five VOCs followed, and they are acetone; benzene; 1-ethenyl-4-methoxy-; ethanone, 1-(2-furanyl)-, together with two esters, butanoic acid hexyl ester and propanoic acid; and pentyl ester.

Microbiota taxa distribution and richness were indicative of a statistically significant change in the alpha diversity estimate between before- and after-dietary-intervention samples, as detected by computing the Faith’s PD metric (Appendix A).

On the contrary, no statistically significant change in alpha diversity resulted when the stratification based on intestinal permeability was used. At the same, when we inspected the beta diversity, neither the treatment nor the intestinal permeability group’s belongings were useful in clustering samples.

Looking for changes in taxa presence/abundance as a consequence of VLCKD treatment, we inspected statistical significance at the genus and species levels, but the applied pairwise statistical test used (Welch BH-corrected test) takes advantage of the taxa collapsing at the genus level (Figure 2). Regardless of the permeability subgroup, when VLCKD-treated (T2) and -untreated (T1) samples were compared, five genera significantly decreased in their relative abundances (Storey-corrected Welch test) after VLCKD administration, whereas five increased (Appendix A). More specifically, *Agathobacter*, *Ruminococcus*, *Fusicatenibacter*, *Collinsella*, and *Bifidobacterium* were all decreased because of dietary treatment administration. On the contrary, the after-treatment increased-genera list included *Streptococcus*, *Bacteroides*, *Eubacterium eligens group*, *Adlercreutzia*, and *UBA1819* (*Ruminococcaceae* family).

We then investigated our sample cohort by using the intestinal permeability as metadata for sample stratification (Figure 2).

Only three genera had relative abundances that differed between post- and pre-altered groups (*Agathobacter*, *Ruminococcus*, and *Bifidobacterium*—Figure 2A), whereas five distinguished post- vs. pre-normal samples (*Agathobacter*, *Roseburia*, *Subdoligranulum*, and *Bifidobacterium—*Figure 2B). Thus, the only non-redundant genus that exclusively marked samples from subjects with altered permeability was *Ruminococcus* (Figure 2A).

### 3.2. Biochemical Pathway Prediction Based on 16S Taxa Abundances

The complete set of taxa (at all taxonomic levels) was used to predict the possible metabolic biochemical pathway profile.

A fold-change analysis of post-altered vs. pre-altered groups highlighted a total of twenty-four pathway among which only two (Bifidobacterium shunt and heterolactic fermentation) resulted in being downregulated after VLCKD treatment (Figure 3). As expected, the list of upregulated variables included seven biochemical pathways related to lipid metabolism: lipid IVA biosynthesis, Kdo transfer to lipid IVA III, palmitate biosynthesis II, oleate biosynthesis IV, super-pathway of fatty acid biosynthesis initiation (*E. coli*), palmitoleate biosynthesis I (from (5Z)-dodec-5-enoate), and stearate biosynthesis II. In the light of a more complex biochemical alteration of microbiota metabolism, another pathway related to biotin and mannan degradation by bacteria, the biopolymer of mannose, emerged (Figure 3 and Table 1).

It is noteworthy that, when post- and pre-VLCKD-administered patients with normal permeability were statistically compared, only two downregulated pathways emerged, i.e., the super-pathway of GDP mannose-derived O antigen building blocks biosynthesis and mannan degradation (Appendix A).

### 3.3. 16S Metabolic Predicted Pathways vs. Untargeted Metabolomics VOCs

With the aim of discovering the significant associations between VOCs and predicted pathways, we used the output of two separately run fold-change analyses. The network plot reported all the statistically significant linear cross-correlations computed based on Picrust2 pathways and fecal/urinary VOCs derived from comparing post-altered and pre-altered sample groups (Figure 4).

The Pearson’s correlation analysis resulted in few statistically significant correlations (r > 0.7) emerging among all cross-linear comparisons. Specifically, the compound hexanal from urines was positively correlated with the *Bifidobacterium* shunt pathway. It is noteworthy that other statistically significant correlations involved the anethole and a list of biochemical pathways, namely mannan degradation, lipid IVA biosynthesis, Kdo transfer to lipid IVA III, ADP-L-glycero-beta-D-manno-heptose biosynthesis, super-pathway of pyridoxal 5-phosphate biosynthesis and salvage and pyridoxal 5-phosphate biosynthesis I, super-pathway of L-methionine biosynthesis via sulfhydrylation, sulfate reduction I assimilatory, and super-pathway of sulfate assimilation and cysteine biosynthesis.

## 4. Discussion

VLCKD treatment leads to several changes in host metabolism regulation, as evidently supported by the volatile organic compounds (VOCs) profile from feces and urines [8], as well as by the Lac/Man ratio [5]. To investigate the commensal gut microbiota taxa, we inspected the 16S rRNA gene sequencing in the presence of normal and altered intestinal permeability. In addition, we connected the related predicted metabolisms with statistically significant VOCs.

Regarding microbiota richness, the dietary intervention leads to an increase in the alpha diversity value, regardless of the intestinal permeability type. It is worth noting that this difference was dissolved when the stratification considered the permeability subgrouping, reflecting a lower power of PLS-DA variables.

On the other hand, the concomitant use of VOCs and microbiota variables from 16S sequencing allowed us to obtain a good separation of sample clusters based on dietary treatment and intestinal permeability, whereas the exploitation of taxonomic annotations alone has a lower power in explaining sample clusters.

With reference to our microbiota data, although obesity has been classically associated with a higher *Firmicutes/Bacteroidetes* ratio, recent analyses did not report a change in this respect and even suggest a heterogeneous taxa association [9,10]. The *Bacteroides*-to-*Firmicutes* ratio has been demonstrated as an erroneous conception to provide any association with health status [9,11,12,13].

The administration of energy-restricted diets rearranges obese and overweight gut microbiota, resulting in an increased microbial diversity and gene richness [14]. We sought to investigate the combination of VLCKD and intestinal permeability by comparing obese patients before and after VLCKD dietary intervention.

By inspecting PLS-DA VIP scores, among the most contributing gut microbial taxa useful in discriminating the four analyzed groups, we identified the butyrate-producing bacterium genus *Butyricimonas*, that ranked first. This genus has been positively associated with weight loss and BMI reduction [15], and we found its increase in both post-treatment groups. Moreover, the exploration of taxa data revealed a decrease in the relative abundance of *Ruminococcus*, and this decrease, in fact, is a hallmark of post-treated altered permeability patients. This genus has been positively correlated with the intake of protein, monounsaturated fat, vitamin A, and vitamin D, and its reduced relative abundance, together with that of *Bifidobacterium*, is indicative of a heightened risk of CVDs’ shared comorbidities with obesity [16].

The presence of other taxa emerging from our statistical analysis is in line with and supported by the recent literature ascertaining how the gut microbiota of frail older people changes in relation to the age of subjects and consequently to the intestinal permeability [17]. This is the case of *Roseburia*, which plays a major role in maintaining the intestinal barrier function and immune defense [18,19], and together with the *Ruminococcus* genus, it exerts a degrading activity against resistant starch and cellulose in plant-based foods, resulting in SCFA production [20]. We measured SCFA by GC-MS target analyses in our previous paper [6], and the only significant difference was found in the butanoic acid level, which decreased after treatment. These two paired results agree on highlighting a detrimental effect of altered permeability in combination with the VLCKD diet, moving the symptom picture far away from gut homeostasis.

The distribution and abundance of microbiota taxa in the intestinal districts are sensitive to various factors, including gastric acid secretion, gastrointestinal peristalsis, and IgA secretion [21], and thus an increase in bacterial fermentative activity may be in turn linked to these factors.

At the same, the diet regimen marked by a disproportion in fat, protein, and fiber produces metabolic waste products, leading to the overgrowth of putrefactive bacteria. This metabolic condition is evidently sustained by higher levels of the urinary marker indican that was found in our obese patient cohort [5]. The increase in specific bacteria like *Bacteroides*, generally resulting from rich-in-fat and poor-in-fiber diets, has been strongly linked to putrefactive dysbiosis [22]. We found a significant increase in *Bacteroides* after treatment in obese patients with normal intestinal permeability but not in those with altered intestinal permeability.

The microbial metabolic-pathway predictions based on sequencing data evidenced a set of statistically significant differences between samples before and after the treatment and allowed us to distinguish between groups based on the intestinal permeability.

The applied approach, although predictive, highlights the downregulation of two fermentation pathways associated with SCFA production, i.e., the “unique shunt of hexose catabolism in *Bifidobacterium*”, which produces primarily acetate and lactate, and the “heterolactic fermentation”. In a linear correlation between non-normally distributed data (fecal/urinary VOCs vs. predicted pathways), the *Bifidobacterium* shunt positively correlated with the decreased levels of hexanal content in urines from patients with altered intestinal permeability. As a derivative from lipid peroxidation and from the decomposition of linoleic acid, this aldehyde is a marker of oxidative stress. Its decrease could be associated with the antioxidant and anti-inflammatory effect exerted by the ketogenic diet [23].

In line with the reverse association between obesity and bifidobacteria observed by Waldram and colleagues [24], we found that the *Bifidobacterium* genus decreased after treatment independently from intestinal permeability.

Among all the other significant upregulated metabolic predictions, two pathways related to biotin metabolism were increased in the gut microbiota of obese patients who underwent VLCKD (8-amino-7-oxononanoate biosynthesis I and biotin biosynthesis I). Recently, the big MetaCardis cohort studies composed of 1500 adult subjects revealed a decrease in biotin biosynthesis and uptake genes and a concomitant reduction in the microbial metabolism of biotin in participants with severe obesity, together with suboptimal circulating biotin levels [25]. Since our obese patients were marked by an increase in biotin metabolisms after dietary treatment, we can consider this shift to be a potential restoration signal towards the homeostasis status.

The peculiar upregulation found for the predicted anhydromuropeptide recycling pathway is indicative of the immune system’s engagement during the chronic inflammation status, a condition that could lead to insulin resistance in obese patients. As well supported by the inspection of microbiota in NOD2-deficient mice, an intact NOD2 peptidoglycan-sensing system is actively involved in metabolic inflammation and insulin resistance and counteracts excessive dysbiosis-linked inflammation and insulin resistance [26].

In VLCKD-treated patients with altered permeability, another group of microbial predicted pathways belonging to fatty acid biosynthesis highlighted the upregulation of metabolisms involved in utilizing palmitoyl-[acp], the precursor of palmitate. More in detail, in this group, the increase in palmitate biosynthesis II, stearate biosynthesis II, palmitoleate biosynthesis I (from (5Z)-dodec-5-enoate), and mycolate biosynthesis led us to argue how the *Bacteroidetes* phylum also produces sphingolipids through bacterial serine-palmitoyl transferase [27]. The activation of this pathway confers resistance to oxidative stress [28], and bacteria-derived sphingolipids modulate host immune responses in the human gut [29].

Recently, a machine learning approach identified the fecal microbial metabolic super-pathway of L-methionine biosynthesis (by sulfhydrylation) as being among the best predictors of obesity status [30]. We found how the VLCKD diet led to increased sulfur amino acid microbial metabolism in the presence of altered intestinal permeability. Bacteria transform sulfate into hydrogen sulfide, which is, in turn, used for cysteine and methionine biosynthesis, i.e., via the super-pathway of sulfate assimilation and cysteine biosynthesis and the sulfate-reduction (assimilatory) pathway [31]. In this context, the chronic intestinal inflammation is the result of hydrogen sulfide impacting the intestinal epithelium at higher concentrations, a multifaceted scenario in which sulfate-reducing bacteria (SRB) contribute to the activation of the immune response [32,33]. Also, the overexpression of lipopolysaccharide biosynthesis-related pathways evidently supports the inflammatory response in altered intestinal permeability. The lipid IVA biosynthesis pathway and the Kdo transfer to lipid IVA III are responsible for the synthesis and functioning of Lipid A, the hydrophobic anchor of bacterial lipopolysaccharide that acts as the elicitor of the eukaryotic innate immune response, acting on pro-inflammatory cytokines [34].

Moreover, the upregulated super-pathway of “GDP-mannose-derived O-antigen building blocks biosynthesis” and “ADP-L-glycero-beta-D-manno-heptose biosynthesis” are crucial in the formation of 6-deoxyhexoses and heptose sugars, which are used as components of the O-antigen of the microbial LPS. A recent study exploring the gut microbiota of an obese Chinese children cohort found an increase in “GDP-mannose biosynthesis” and “UDP-N-acetylglucosamine-derived O-antigen building blocks biosynthesis” pathways in metabolic healthy obese subjects compared with a metabolically unhealthy obese cohort [35]. Since the GDP-mannose pathway is statistically significant but divergent in the fold-change analyses of both our subgroups (increased in altered samples and decreased in normal permeability samples), the effectiveness of VLCKD treatment is directly linked to the intestinal permeability status.

Our data revealed how the microbial pathway responsible for the chondroitin sulfate degradation significantly increased after VLCKD in altered permeability subjects. These data have to be evaluated concomitantly to the increase in *Bacteroides* genus that arose from our 16S analysis. Indeed, the literature evidence reported a link between chondroitin sulfate and the increase in the relative abundance of the gut bacterial genus *Bacteroides* [36].

Studies on HFD-administered mice demonstrate how oligosaccharides have a lipase-inhibitory activity impacting the triglycerides absorption in the intestine and have an adipocyte-inhibitory activity to reduce lipid accumulation [37]. Looking for an association between microbial taxa and visceral fat accumulation, Nie and colleagues applied a metagenomics approach that is useful in discriminating the most enriched microbial pathways in obesity. The 10 pathways responsible for energy generation included the D-glucarate degradation I pathway [38]. Accordingly, we detected a significant increase in the microbial D-glucarate degradation I pathway after VLCKD treatment with altered permeability. Two other important upregulated pathways (pyridoxal 5-phosphate biosynthesis and the super-pathway of pyridoxal 5-phosphate biosynthesis and salvage) in altered permeability samples indicated the role of pyridoxal 5-phosphate biosynthesis. Pyridoxine increases fat oxidation and significantly improves insulin sensitivity in overweight and obese subjects [39] by attenuating Ca^2+^ influx, which stimulates fatty acid synthase expression and activity.

The panel of statistically significant metabolites includes compounds that are not derivative from microbial metabolism. The increased level of anethole, an aromatic compound derived from fennel and anise essential oils, is linked with antioxidant and anti-inflammatory effects and positively correlates with important energetic, lipid, and fat oxidation pathways. Importantly, this molecule impacts energy expenditure by inducing thermogenesis in adipocytes and muscle cells in obese patients [40].

Finally, we detected an increase in the pathway of heme biosynthesis, and a recent paper argued about a possible role of heme biosynthesis in rat and human adipogenesis [41]. Based on the study on heme biosynthesis-related gene expression, the postulated hypothesis is that heme allows us to achieve an optimal adipocyte differentiation by sustaining mitochondrial function.

As a limitation of this study, although our results are supported by the literature evidence, it is worth mentioning that 16S sequencing output has limited resolution at species and genus levels and may suffer from statistical errors. It was downstream, inferring pathway prediction is not comparable with shotgun sequencing-derived data (metagenomics). A new experimental design based on metagenomic and metatranscriptomics sequencing is being planned.

## 5. Conclusions

Our experimental design aimed at investigating existing differences in microbiota taxa and volatile organic compounds in order to compare a cohort composed of obese patients, marked by altered and normal intestinal permeability, before and after a VLCKD treatment.

A metabolic prediction of biochemical pathways, derived from microbiota taxa relative abundances, was of aid in identifying a shift in the intestinal homeostasis status and in tracing the connection with latent inflammation phenotype.

Although suffering from a statistical bias, predicted bacterial metabolisms have been reported and discussed in the recent literature papers dealing with obese patients administered restrictive dietary regimens.

Taken together, these data confirm and extend the knowledge of specific processes involved in obesity comorbidities, which ultimately depend on immune system activation and on an altered intestinal permeability status.

## Figures and Tables

**Figure 1 nutrients-16-02079-f001:**
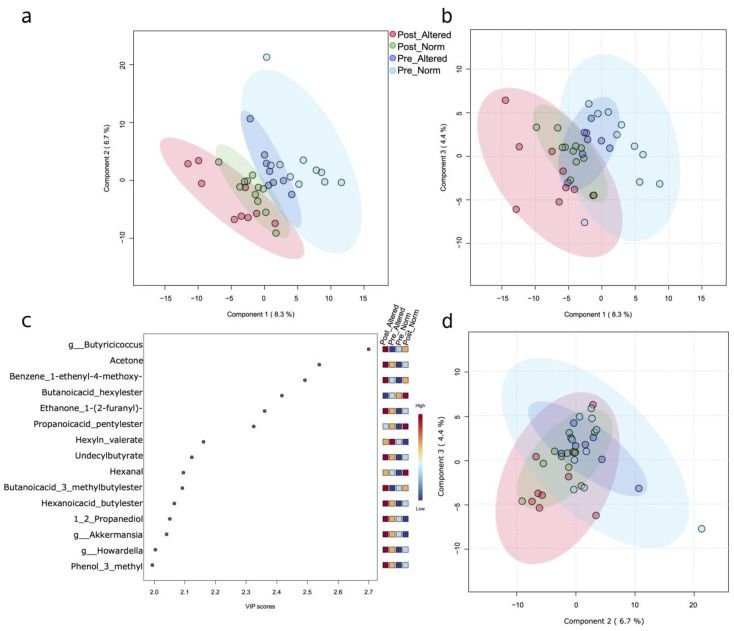
PLS-DA plot. A normalized matrix composed of fecal and urine metabolites, plus 16S taxa, was used as input for the PLS regression analysis. (**a**) Component 1 versus component 2 PLS-DA score plot based on 16S metataxonomic and metabolomics (fecal and urinary VOCs) variables. (**b**,**d**) Relative to other component combinations and, precisely, component 1 vs. 3 and component 2 vs. 3, respectively. (**c**) Top fifteen “Variable Importance in Projection” (VIP) scores, including both VOCs and microbiome taxa with a value greater than 2.0.

**Figure 2 nutrients-16-02079-f002:**
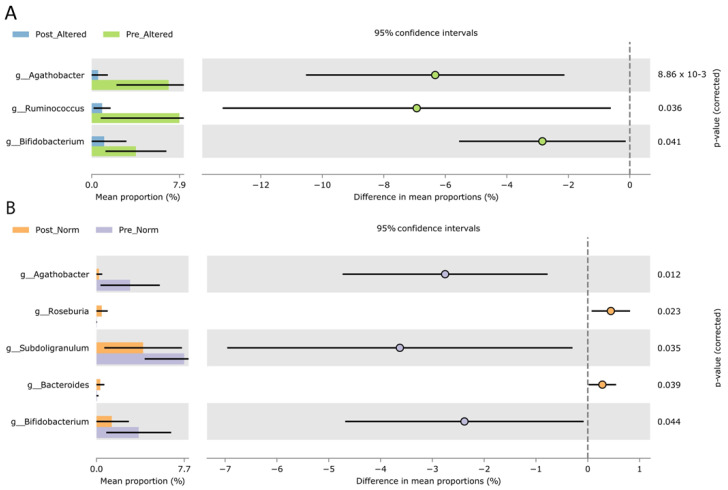
Extended error bar plots of statistically significant bacterial genera that differed as a consequence of VLCKD dietary treatment in post-altered versus pre-altered samples (**A**) and post-versus pre-normal samples (**B**). Both the plots report the effect size and associated confidence interval for each significative feature (corrected *p* < 0.05) obtained in STAMP software by applying corrected Welch test (BH) statistics based on the normalized matrix from QIIME2. Difference in genus mean proportions (95% of confidence intervals) resulted from a Welch pairwise test (corrected by applying a Benjamini–Hochberg multiple test). Post-treatment groups are colored light blue (post-altered) or orange (post normal), and green (pre-altered) or violet (pre-norm). All the reported genera were statistically significant after correction (corrected *p* < 0.05). Because of the direction of the comparison, the differences in mean proportion for pre-altered samples appeared as negative values.

**Figure 3 nutrients-16-02079-f003:**
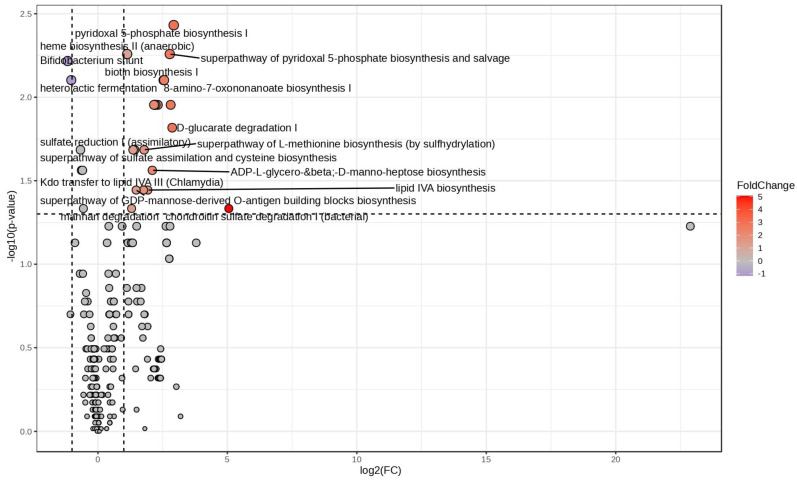
Log2 fold change vs. significance plot. Statistically significant Picrust2 pathways obtained by comparing post-altered and pre-altered sample groups visualized as a volcano plot. Fold change is indicative of increasing (red) and decreasing (violet) abundance in post-altered predicted pathways.

**Figure 4 nutrients-16-02079-f004:**
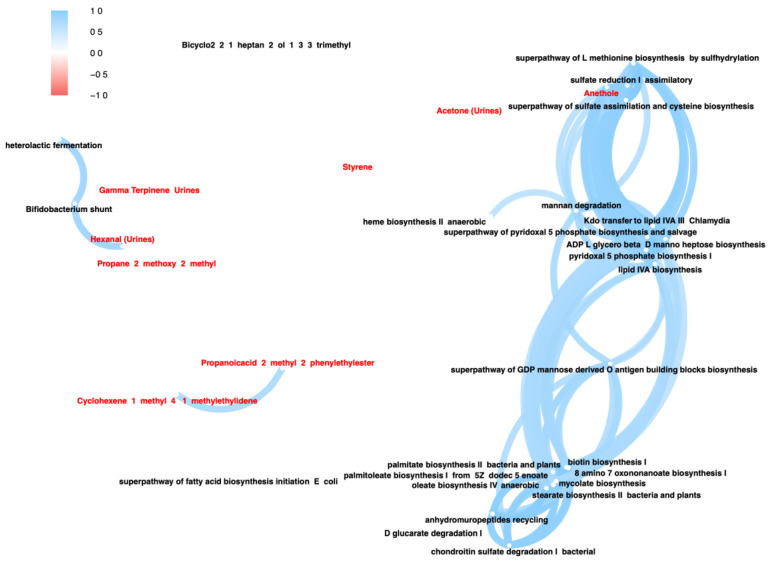
Pearson correlation between statistically significant predicted pathways and fecal/urinary VOCs, as derived from the comparison between pre- and post-VLCKD-administered subjects with altered intestinal permeability. Network plot of significant linear correlations reported as linear connection (*p*-value < 0.05 and r > 0.7). Blue lines indicate positive correlations among variables (nodes). Red font indicates fecal and urinary VOCs, whereas black font is relative to metabolic pathways.

**Table 1 nutrients-16-02079-t001:** Statistically significant Picrust2 up- and downregulated pathway table. Each down- or upregulated pathway (in altered-permeability VLCKD-treated subjects due to the direction of the pairwise comparison) is related to statistical descriptors, including fold change (FC), log2(FC), significance (raw.pval), and -LOG10(p).

Pathway	FC	log2(FC)	Corrected *p*-Value	−LOG10(p)
Chondroitin sulfate degradation I (bacterial)	33.248	5.0552	0.046401	1.3335
Pyridoxal 5-phosphate biosynthesis I	7.6054	2.927	0.0037022	2.4315
D-glucarate degradation I	7.3074	2.8694	0.01522	1.8176
Anhydromuropeptides recycling	7.0071	2.8088	0.011107	1.9544
Super-pathway of pyridoxal 5-phosphate biosynthesis and salvage	6.8423	2.7745	0.0055121	2.2587
8-amino-7-oxononanoate biosynthesis I	5.8769	2.555	0.007898	2.1025
Biotin biosynthesis I	5.7884	2.5332	0.007898	2.1025
Palmitate biosynthesis II (bacteria and plants)	4.9619	2.3109	0.011107	1.9544
Oleate biosynthesis IV (anaerobic)	4.6762	2.2253	0.011107	1.9544
Super-pathway of fatty acid biosynthesis initiation (E. coli)	4.596	2.2004	0.011107	1.9544
Palmitoleate biosynthesis I (from (5Z)-dodec-5-enoate)	4.5835	2.1965	0.011107	1.9544
Stearate biosynthesis II (bacteria and plants)	4.5657	2.1908	0.011107	1.9544
mycolate biosynthesis	4.4898	2.1666	0.011107	1.9544
ADP-L-glycero-beta-D-manno-heptose biosynthesis	4.2874	2.1001	0.027396	1.5623
lipid IVA biosynthesis	3.8011	1.9264	0.035952	1.4443
Super-pathway of L-methionine biosynthesis (by sulfhydrylation)	3.4462	1.785	0.02065	1.6851
Kdo transfer to lipid IVA III (Chlamydia)	3.3877	1.7603	0.035952	1.4443
Super-pathway of GDP-mannose-derived O-antigen building blocks biosynthesis	2.7817	1.476	0.035952	1.4443
Sulfate reduction I (assimilatory)	2.6706	1.4172	0.02065	1.6851
Super-pathway of sulfate assimilation and cysteine biosynthesis	2.5613	1.3569	0.02065	1.6851
Mannan degradation	2.4624	1.3001	0.046401	1.3335
Heme biosynthesis II (anaerobic)	2.1869	1.1289	0.0055121	2.2587
Heterolactic fermentation	0.48974	−1.0299	0.007898	2.1025
Bifidobacterium shunt	0.44482	−1.1687	0.0060678	2.217

## Data Availability

The datasets used and/or analyzed during the current study are available from the corresponding author upon reasonable request. The data are not publicly available due to privacy.

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
