# Peer review of "A Multi-Omics Approach to Disclose Metabolic Pathways Impacting Intestinal Permeability in Obese Patients Undergoing Very Low Calorie Ketogenic Diet"

_nutrients, 2024, doi:10.3390/nu16132079_

Round 1
Reviewer 1 Report
Comments and Suggestions for Authors
Celano and coworkers present a multiomic assessment of a dietary intervention on obese subjects using a low-calorie and ketogenic diet. The use of such nutritional regime in obesity is not new; however, stratification of patients according to intestinal barrier deterioration is intriguing. This peer-reviewer has raised the following concerns, making this study unsuitable for publication in its current form:
1. First of all, I encourage the authors to do a professional, and scientific proofreading of the manuscript as it shows a lot of typos across the document (e.g. “Kruscal”), and they should improve the technical aspects of the assessment. There is a wide variety of terms referring to a sole approach used in the study, leaving the impression to non-specialist readers and the general audience of being talking about different things and allowing confusion; for instance, "16S microbiota profile", "16S abundance", "16S metagenomic sequencing", "metabarcoding", "metataxonomics", "16S rRNA gene sequencing", “16S RNA analyses”, “16S taxa abundance”. The 16S rRNA gene amplicon sequencing approach IS NOT metagenomics, and the proper terminology must be used for such aims (doi: 10.1186/s40168-015-0094-5, doi: 10.1186/s40168-020-00875-0). Similarly, the utilisation of the terms "untargeted metabolomics", "VOCs", "volatilome", "metabolomics", and "GC-MS volatilome" does not help the reader understand in an unequivocal manner what the authors have done precisely. So, a uniform terminology and accepted nomenclature must be adopted in both cases.
2. An explanation of patient dropout must be presented as a flow chart following CONSORT guidelines (item#13) recommendations. The methods describe recruiting 25 patients (50 sample points), but in faecal DNA extractions for microbiota assessment, only use 40 samples (20 subjects), and in Schema 3, it is declared the use of 38 samples (19 subjects).
3. The microbiome data is compositional; therefore, an appropriate transformation (e.g., the CLR method) should be applied accordingly to minimize the number of false positive categories resulting from group comparison (check doi: 10.3389/fmicb.2017.02224). Despite declaring to having used the QIIME2 suite of analysis, I'm afraid the authors have not followed the recommended guidelines to deal with the sparse and compositional nature of microbiota data. This is intuited from scaling used to measure the abundance of bacterial taxa (Figure 2, relative abundance).
4. The colour legend of scatter points is not seen in any of the "Schema" presented.
5. The human-eye cannot easily see and intuit the information compiled in three-dimensional plots, as presented in Figure 1. I recommend plotting bi-dimensional plots across all different component combinations (N = 3).
6. The chronological order to present multivariate assessment from microbiota and metabolomics, individually, should precede the exploration in a combined manner (subheadings 3.1 and 3.2).
7. Functional analysis based on amplicon information with very limited resolution at the species level, even at the genus level, according to the results described, and using the PICRUSt tool is worthless, in my opinion. More reliable assessments must be done using shotgun sequencing-derived data (metagenomics). Moreover, the kind of integrative analysis (by using simple linear correlations on non-normally distributed data) has a poor rationale and it is not properly explained during discussion. The apparent link between host metabolites and bacterial metabolic pathways should be better argued to establish solid and coherent relationships.
8 The Bacteroides-to-Firmicutes ratio is an erroneous conception to provide any association with health status (doi: 10.3390/nu12051474, doi: 10.1038/s41467-020-18871-1, doi: 10.1128/mBio.01018-16, doi: 10.1016/j.febslet.2014.09.039). Please avoid using such a metric to make health-associated assumptions, which means nothing.
9. I'm afraid I have to disagree with the continued utilisation of the term "dysbiosis". As the authors should be aware, the definition of such this terminology is a matter of debate; therefore, its generic utilisation should be avoided (doi: 10.1128/mBio.01492-17, doi: 10.1111/1751-7915.13479, 10.1038/nmicrobiol.2016.228). Besides, the authors should be additional explain what’s the “fermentative” and “putrefactive” dysbiosis.
10. The authors must be coherent in granting beneficial and harmful roles to concrete bacterial groups. The reader can be confused if in one part of the text "Bacteroides" appear as harmful entities and then you grant a "beneficial role" later on (lines 371-373 vs 431-434).
11. Table 1 must be arranged by decreasing FC to better prioritize candidates.
12. Conclusions must be re-written as the general aspects of the intervention. I'm afraid this particular dietary regimen is not beneficial at all for the patients being subjects of this clinical trial, and this is not perceived in such statement.
Comments on the Quality of English Language
I encourage the authors to do professional and scientific proofreading of the manuscript, as it contains many typos.
Reviewer 2 Report
Comments and Suggestions for Authors
Celano et al. present a study whereby the metabolomics of patients on a VLCKD is presented. The work appears premature, with author’s missing and unclear references, along with no supplemental images presented (although they are described). Further, much of the methods remain poorly described, along with the Figure’s being significantly reduced in quality/size to be unreadable. As such, while the work may hold value, it’s quite challenging to determine that in it’s given form.
Minor:
1. Missing author as the author list ends with “and*”
2. Spelling error in abstract “disfunctions”
3. Line 52 - Spelling error in introduction “signalling”
4. Line 141, is the (q < 0.05) supposed to be an “a” or a “p” or “corrected p”, as defined in the test later.
5. Section 2.6 references “our previous paper(8)”. None of the authors in the referenced paper appear to be authors on the current paper.
6. Scheme 1 and Scheme 2 are entirely unreadable at any magnification.
7. Figure 1, component 2 and component 3 is covering the axis.
8. Figurer 1 has none of the indices discussed “blue for pre-altered, red for post-altered, green for post-norm, aqua for pre-norm”. Also, aqua versus blue isn’t ideal, they are too close in color.
9. As understood from the materials and methods, there are two subclusters, one with 14 samples and normal permeability and one with 11 samples with altered permeability. But fecal samples were collected before and after the VLCKD in only 20 patients, from which group of permeability was the fecal sample collected? When was the before sample collected? The day before the diet started or before the supplements and vitamins were stopped?
10. The term VOC is never defined.
11. Scheme 3 mentions a VLCKD treated versus not-administered group but the methods do not mention a control group.
12. Figure 2 is unreadable.
13. The supplemental figures aren’t available.
14. Figure 3 is presented with words in the figure and it’s difficult to determine what data point is correlated to those words.
15. None of Figure 4 is remotely readable.
Comments on the Quality of English LanguageEnglish is fine
Reviewer 3 Report
Comments and Suggestions for Authors
In the manuscript submitted to me for review entitled "Very Low-Calorie Ketogenic Diet in obese patients: a multi-omics approach to disclose the impact of altered intestinal permeability“ the authors Giuseppe Celano, Francesco Maria Calabrese, Giuseppe Riezzo, Benedetta D'attoma, Antonia Ignazzi, Martina Di Chito, Annamaria Sila, Sara De Nucci, Roberta Rinaldi, Michele Linsalata, Carmen Aurora Apa, Maria De Angelis, Gianluigi Giannelli, Giovanni De Pergola and Francesco Russo investigated the 16S micro-profile of the gut microbiota in obese patients with normal or altered intestinal permeability before and after administration of a Very Low-Calorie Ketogenic Diet (VLCKD).
The study included patients aged 18 - 65 years, and their participation in the investigation was in accordance with the guidelines of the Declaration of Helsinki and approved by the IRCCS "S. de Bellis” Scientific Committee and the Institutional Ethics Committee of IRCCS “Ospedale Oncologico—Istituto Tumori Giovanni Paolo II”. The consent of each patient for his participation in the research was obtained beforehand.
To support their research, the authors used 36 references that presented information from studies published mostly in the past decade. Almost 3/4 of the total references are from the last 5 years, with nearly 1/3 of them from the current year 2024. This shows that the topic under consideration is relatively new and up to date and would be of interest to Nutrients readers. I did not notice any redundant self-citations, all references used are appropriate and necessary for the preparation of the manuscript.
My remarks and recommendations to the authors are:
I have no objections to the chosen methods and the conduct of the study itself. My remarks are mainly related to the technical way of presenting the results.
1. The inscriptions on the abscissas and ordinates, as well as inside the figures and schemes themselves, are too small and cannot be read. The font needs to be increased - most figures and schemes are large enough and this should not be a problem. An exception is only figure 1, in which the font of the inscriptions is fine.
2. In Scheme 2, in addition to the above problem with the inscriptions, on the right side, the limiting line at the top edge is thinner than the others. Let it be done with the size of the rest of the lines to make the whole scheme look better.
3. 5 supplementary figures are described in the text and in the back section of the manuscript. But the supplementary file is not attached. Let it apply.

Round 2
Reviewer 2 Report
Comments and Suggestions for Authors
This work is a resubmission where the author’s haven’t been really responsive to revision. The work still has the methods very poorly described and many of the Figure’s remain unreadable or not relevant to the overall hypothesis. Further, this is the 4th paper in less than a year that has been published by this group using this same experimental cohort. All of them have roughly the same title involving “obese patients on a very-low ketogenic diet” and “metabolomics” and “intestinal barrier and/or permeability”. It’s not immediately clear what the difference is between the 4 published works and some of the data does look as if it’s overlapping. In the discussion, some of the sentences are copied almost verbatim from one of the other paper’s, ex; “The aldehyde hexanal, a derivative of lipid peroxidation and a decomposition product of linoleic acid, has also been recognized to be a marker of oxidative stress, and its decrease could be associated with the antioxidant and anti-inflammatory effect exerted by the ketogenic diet”. Further, there needs to be consistency between the nomenclature. It is unclear from the legends which graphs describe these two groups (pre versus post and altered versus norm). It is recommended that legends and graphs contain an obese control versus an obese impaired permeability group for clarity.
The emphasis of this work is supposedly the separation of an obese group on a VLCKD into two groups with differences in intestinal permeability. How was intestinal permeability assessed between the two groups? This data needs to be included in this work. It is stated in the discussion (first paragraph) that “VLCKD treatment leads to several changes in host metabolism regulation, as evidently supported by the volatile organic compounds (VOCs) profile from feces and urines, as well as by markers including the Lac/Man ratio, indican, skatole, and lipopolysaccharide (LPS) [8]. To investigate the commensal gut microbiota taxa, we inspected the 16S rRNA gene sequencing in the presence of normal and altered intestinal permeability”. The implication here is that reference 8 will clearly show differences in Lac/Man ratio, indican, skatole, and lipopolysaccharide between the two obese groups on VLCKD, providing insight on how the two groups were separated. This is not the case, as the reference shows these are variables that contribute to intestinal permeability. However, in the published article “A Pilot study” these variables are measured. It’s mentioned in this paper that a subgroup (Figure 4) had particularly elevated Lac/Man ratio, but as far as the reviewer can tell, these were never separated out statistically but rather with a cutoff value of 0.03. Of the other indices mentioned, indican, skatole, and LPS, these are in Figure 6 of the prior work but note that they are lumped together into one group with indican and LPS rising significantly in all the obese groups. As such, it’s unclear if these variables that define the two groups are actually different between them and this data needs to be added to the work here.
Minor Thoughts:
1. The inset in Figure 1A is pretty hard to read, especially the legend. I don’t have an easy solution to this particular problem, given the space constraints. Maybe consider adding it to the supplemental data just by itself? Also, the legend says “in the Panel A right upper quarter”, do you mean, “upper left quarter”? Also, I think Panel 1C and 1D are switched in the figure.
2. Line 200, “Although with a higher….” is an incomplete sentence.
3. Supplementary Figure S3 should probably be renamed to “Before Treatment” and “After Treatment” as opposed to Group 1 and Group 2.
4. Figure 2 and S4 remain challenging to interpret. For instance, in Figure 2, it looks like the normalization is placed against the pre-altered state, meaning that Agathobacter decreased and thus is represented by a negative number. However, in Figure S4, the X axis is now placed against the post-normal state, meaning that the X-axes is now a positive number despite it decreasing. This is not a good representation and is likely to confuse readers. Further, there isn’t a good definition of what the author’s mean by altered versus normal and why these are significant, or not.
5. Figure 4A – what exactly is the emphasis or point of figure 4? What is currently there is black and red letters (unreadable unless at 200% magnification) and blue squiggles that are all the same color. There needs to be a better way of interpreting and presenting this data.
There are some figures that mention urinary VOC and some that do not. The methods only mention fecal extraction. How were the urinary and fecal VOC’s distributed and defined in each figures and analysis?
